# TabGraphs: new benchmark and insights for learning on graphs with tabular features

## Abstract

The field of tabular machine learning is very important for industry and science. Table rows are typically treated as independent data samples. However, often additional information about the relations between these samples is available, and leveraging this information may improve the predictive performance. As such relational information can be naturally modeled with a graph, the field of tabular machine learning can borrow methods from graph machine learning. However, graph models are typically evaluated on datasets with homogeneous features, such as word embeddings or bag-of-words representations, which have little in common with the heterogeneous mixture of numerical and categorical features distinctive for tabular data. Thus, there is a critical difference between the data used in tabular and graph machine learning studies, which does not allow us to understand how successfully graph methods can be transferred to tabular data. In this work, we aim to bridge this gap. First, we create a benchmark of diverse graphs with heterogeneous tabular node features and realistic prediction tasks. Further, we evaluate a vast set of methods on this benchmark, analyze their performance, and provide insights and tips for researchers and practitioners in both tabular and graph machine learning fields.

## 1 Introduction

Tabular data is ubiquitous in industry and science, thus machine learning algorithms for tabular data are of great importance. A key distinction of tabular data is that it typically consists of a mixture of numerical and categorical features, with numerical features having vastly different scales and distributions, and all features having different meaning and importance for the task. We further call such features *heterogeneous*. Deep learning methods often do not work well with heterogeneous features, thus, the machine learning methods of choice for tabular data are decision-tree-based models, in particular gradient-boosted decision trees (GBDT) (Friedman, 2001). However, there is a growing number of recent works trying to adapt deep learning methods to tabular data (Arik & Pfister, 2019; Badirli et al., 2020; Huang et al., 2020; Gorishniy et al., 2021).

In tabular machine learning, table rows are typically treated as independent data samples. However, often there is additional information available about relations between these samples, and leveraging this information has the potential to improve predictive performance. Such relational information can be modeled with a graph. There are many areas where graphs naturally arise. For example, if you have an application where users can interact with each other in some way, you essentially have a graph. Examples of this include social networks, discussion forums, question-answering websites, chat applications, financial transaction networks. Even if users do not directly interact with each other, meaningful relations can often be created, for example between users buying similar products on shopping websites, users watching similar content on video hostings, and workers doing the same tasks on crowd-sourcing platforms. In all these and many other cases, using graph information can improve the quality of machine learning model predictions.

Graph machine learning is a field focused on the development of methods for learning on graph-structured data. In last years, the most successful models for such data have become graph neural networks (GNNs) (Kipf & Welling, 2016; Gilmer et al., 2017). Thus, it can be desirable to adapt these models to tabular data. However, GNNs are typically evaluated on graphs with homogeneous features (most frequently bags of words or bags of word embeddings), which are very different from

heterogeneous features present in tabular data, and because of this, it is unclear how they will perform on tabular data. Recently, there have been several methods proposed specifically for learning on graphs with heterogeneous node features (Ivanov & Prokhorenkova, 2021; Chen et al., 2022). However, both of these works note the lack of publicly available graph datasets with heterogeneous features, and thus their evaluation setting is limited. This highlights the difference between industry, where data with heterogeneous features is abundant, and graph machine learning benchmarks, where such data is barely present. We believe that this difference holds back the adoption of graph machine learning methods to tabular data.

In our work, we aim to bridge this gap. First, we create a benchmark of graphs with heterogeneous tabular features — TabGraphs. Our benchmark has realistic prediction tasks and is diverse in data domains, relation types, graph sizes, graph structural properties, and feature distributions. Further, we evaluate a large set of machine learning methods on this benchmark. We consider models for tabular data, both GBDT and deep learning ones, several GNN architectures with different numerical feature pre-processing strategies, and the recently proposed methods for graphs with heterogeneous features. Our main findings are:

- Using the graph structure in data and applying graph machine learning methods can lead to an increase in predictive performance in many real-world datasets.

- The predictive performance of different GNN architectures may vary significantly across datasets, and the best-performing architecture cannot be chosen without experiments.

- Standard GNNs typically outperform recently proposed methods designed specifically for node property prediction tasks on graphs with tabular features.

- Adding numerical feature embeddings that have been proposed in tabular deep learning to GNNs can further improve their performance on graphs with heterogeneous features.

To summarize, our main contributions are as follows:

- We introduce TabGraphs, the first benchmark for learning on graphs with heterogeneous node features, which covers various industrial applications and includes graphs with diverse structural properties and meaningful prediction tasks;

- Using the proposed benchmark, we evaluate a vast set of models, including standard baselines and advanced neural methods for both tabular and graph-structured data, and analyze the results of our experiments;

- Based on our empirical study, we provide insights and tips for researchers and practitioners in both tabular and graph machine learning fields.

We will make our benchmark and the code for reproducing our experiments public once we finalize legal procedures. We also plan to maintain our benchmark and add additional datasets in the future.

## 2 RELATED WORK

**Machine learning for tabular data** The key distinction of tabular data is that it usually consists of a mixture of numerical and categorical features. Standard deep learning models often can not provide decent performance on such features. Thus, the methods of choice for tabular data are typically based on the ensembles of Decision Trees, such as Gradient Boosted Decision Trees (GBDT) (Friedman, 2001) with the most popular implementations being XGBoost (Chen & Guestrin, 2016), LightGBM (Ke et al., 2017), and CatBoost (Prokhorenkova et al., 2018). However, deep learning models have several advantages compared to the tree-based ones, such as modularity and the ability to learn meaningful data representations. Because of that, there has been an increasing number of works trying to adapt deep learning to tabular data (Klambauer et al., 2017; Wang et al., 2017; Arik & Pfister, 2019; Song et al., 2019; Popov et al., 2019; Badirli et al., 2020; Hazimeh et al., 2020; Huang et al., 2020; Gorishniy et al., 2021; Wang et al., 2021; Gorishniy et al., 2022). Among these works, the retrieval-augmented deep learning models (Kossen et al., 2021; Qin et al., 2021; Somepalli et al., 2021; Gorishniy et al., 2023) are particularly relevant. For each data sample, these models retrieve the information about other labeled examples from a database, which is typically

done using some form of attention mechanism (Bahdanau et al., 2014), and use it to make a prediction. Thus, these models learn to find other relevant samples in the dataset, where relevance is determined by feature similarity. In contrast, in our work, we assume that some relations between data samples are already given in advance, which is common in many real-world applications, and focus on the models that can utilize these relations.

**Machine learning for graphs**  Graphs are a natural way to represent data from various domains. Due to this, machine learning on graph-structured data has experienced significant growth in recent years, with Graph Neural Networks (GNNs) showing particularly strong results on many graph machine learning tasks. Numerous GNN architectures have been proposed (Kipf & Welling, 2016; Hamilton et al., 2017; Veličković et al., 2017), and most of them can be unified under a general Message Passing Neural Networks (MPNNs) framework (Gilmer et al., 2017). However, GNNs are typically evaluated on graphs with homogeneous node features, such as bag-of-words or bag-of-word-embeddings representations. Let us review some popular graph datasets for node classification to emphasize this point. The most frequently used datasets are the three citation networks `cora`, `citeseer`, and `pubmed` (Sen et al., 2008; Namata et al., 2012; Yang et al., 2016; McCallum et al., 2000; Giles et al., 1998). The first two datasets use bags-of-words as node features, while the third one uses TF-IDF-weighted bags-of-words. Other datasets for node classification often found in the literature include citation networks `coauthor-cs` and `coauthor-physics` (Shchur et al., 2018) that use bags-of-words as node features, and co-purchasing networks `amazon-computers` and `amazon-photo` (Shchur et al., 2018) that also use bags-of-words. In the popular Open Graph Benchmark (OGB) (Hu et al., 2020) `ogbn-arxiv`, `ogbn-papers100M`, and `ogbn-mag` datasets again use bags-of-words as node features, while `ogbn-products` uses dimensionality-reduced bag-of-words representations. In the recently proposed benchmark of heterophilous graphs (Platonov et al., 2023), `roman-empire` dataset uses word embeddings as node features, while `amazon-ratings` and `questions` datasets use bags-of-words. From these examples, it becomes clear that the effectiveness of graph ML models on graphs with heterogeneous node features remains under-explored.

**Machine learning for graphs with heterogeneous features**  Recently, two methods have been proposed specifically for learning on graphs with heterogeneous tabular features. One of them is BGNN (Ivanov & Prokhorenkova, 2021), an end-to-end trained combination of GBDT and GNN, where GBDT takes node features as input and predicts node representations that are further concatenated with the original node features and used as input to a GNN. Another is EBBS (Chen et al., 2022), which alternates between boosting and graph propagation steps, essentially fitting GBDT to a graph-aware loss, and is also trained end-to-end. However, as both of the works note, there is a lack of publicly available datasets of graphs with heterogeneous features. For this reason, the evaluation of the proposed methods is limited: some of the used graphs do not have heterogeneous features, and some others have various issues which we describe in detail in Appendix B. Thus, better benchmarks for evaluating models for graphs with heterogeneous features are needed.

## 3 TabGraphs: new graph benchmark with tabular features

In this section, we introduce a new benchmark of graph datasets with heterogeneous node features. Some of the datasets are taken or adapted from open sources, while others are obtained from proprietary logs of products and services of a large IT company. Below, we describe the proposed datasets.

### 3.1 Proposed graph datasets

**tolokers-tab**  This is a new version of the dataset `tolokers` from Platonov et al. (2023). It is based on the data from the Toloka crowdsourcing platform (Likhobaba et al., 2023). The nodes represent tolokers (workers) and they are connected by an edge if they have worked on the same task. Thus, the graph is undirected, and the task is to predict which tolokers have been banned in one of the tasks. We kept the original graph and task, but we extended the node features which made them heterogeneous. The new node features are shared with us by the dataset creators and are based on the workers task performance and profile information.

**questions-tab**   This is a new version of the dataset `questions` from Platonov et al. (2023). It is based on the data from a question-answering website. Here, nodes represent users, and two users are connected by an edge, if either of them has answered to the other's question. The resulting graph in undirected, and the task is to predict which users remained active on the website (i.e., were not deleted or blocked). We kept the original graph and task, but extended the node features to make them heterogeneous. The original version of this dataset used BOW embeddings as user features, while we provide a number of attributes based on the user profile information and their activity on the website. Similarly to `tolokers-tab`, the extended features are shared with us by Platonov et al. (2023). Note that these new features are much more predictive of the target, as demonstrated by better performance achieved by models on the new version of the dataset.

**amazon-users**   This is a known academic graph dataset for fraud detection with tabular features that has been introduced in (Zhang et al., 2020). It represents a network of customers with the edges corresponding to some sort of relation between them depending on their reviews under various products at Amazon. The original study proposed to use the relations between users based on their shared products (`upu`), same scores (`usu`), or similar content of their reviews (`uvu`). In this paper, we do not consider graphs with different edge types, so we chose one relation `uvu` as an interesting way to construct a graph that is different from other datasets in our benchmark. The set of features includes the number of rated products, review sentiment, username length, counters of positive and negative ratings, and other statistics based on user activity. The resulting graph is undirected, and the task is to predict whether a user is a fraudster. In our preliminary experiments, we found that one may achieve $\sim 0.99$ ROC-AUC by using a simple graph-agnostic GBDT model, so we decided to remove the number of votes as the most important feature, which also was used for the computation of target variable and thus may have caused an information leak.

**city-reviews**   This dataset is obtained from the logs of a review service. It represents the interactions between users of the service and various organizations located in two major cities. The organizations are visited and rated by users, so the dataset is originally bipartite and contains entities of these two types. Thus, we transform it by projecting to the part of users. Let $\mathbf{P} \in \{0,1\}^{m \times n}$ be a binary adjacency matrix of users and organizations, where $m$ is the number of organizations, $n$ is the number of users, and $p_{ij}$ denotes whether a user $j$ has left a review for an organization $i$. Then, $\mathbf{B} = \mathbf{P}^T\mathbf{P} \in \mathbb{R}^{n \times n}$ corresponds to the weighted adjacency matrix of users, where $b_{ij}$ is the dot product of columns $i$ and $j$ in $\mathbf{P}$. Here, the more common rated organizations there are for two users, the greater the weight of the connection between them. Further, we obtain a binary adjacency matrix $\mathbf{A} \in \{0,1\}^{n \times n}$ of users with $a_{ij} = [b_{ij} \geqslant \gamma]$ by applying a threshold $\gamma = 2$ to the weights $b_{ij}$. The resulting graph is undirected, and the task is to predict whether a user is a fraudster. Most of the node attributes are related to the activity of users at the review service.

**hm-products**   This is an open-source dataset that has been introduced at the Kaggle competition organized by H&M Group (García Ling et al., 2022). This dataset is originally bipartite and contains entities of two types — customers and products that they purchase at the H&M shop. Thus, we transform it by projecting to the part of products. The connections in the original dataset can be described by a weighted adjacency matrix $\mathbf{P} \in \mathbb{R}^{m \times n}$, where $m$ is the number of users, $n$ is the number of products, and $p_{ij}$ denotes how many times a user $i$ has bought a product $j$. Then, $\mathbf{B} = \mathbf{P}^T\mathbf{P} \in \mathbb{R}^{n \times n}$ corresponds to the weighted adjacency matrix of products, where $b_{ij}$ is the dot product of columns $i$ and $j$ in $\mathbf{P}$. The more often either of two products is bought by a common customer, and the more shared customers there are in general, the greater is the weight $b_{ij}$ of the connection between these products. After that, we obtain a binary and more sparse adjacency matrix $\mathbf{A} \in \{0,1\}^{n \times n}$ of products with $a_{ij} = [b_{ij} \geqslant \gamma]$ by applying a threshold $\gamma = 10$ to the weights $b_{ij}$. The resulting graph is undirected. For this dataset, we consider two different versions: `hm-groups` with the product group as the target for the classification task and `hm-prices` with the average price of a product as the target for the regression task. In both cases, we adjust the set of features so that the problem does not become trivial, but the underlying graph is the same for these two versions. For the regression task, we consider such features as product types, graphical appearance, color group names, etc. For the classification task, the set of features includes average price and a reduced subset of categorical attributes from the regression task, which makes the problem more challenging.

Table 1: Statistics of the proposed TabGraphs benchmark.

| | node classification | | | | | node regression | | |
|---|---|---|---|---|---|---|---|---|
| | tolokers-tab | questions-tab | amazon-users | city-reviews | hm-groups | city-roads | avazu-devices | hm-prices |
| # nodes | 11.8K | 48.9K | 11.8K | 148.8K | 46.5K | 159.9K | 76.3K | 46.5K |
| # edges | 519.0K | 153.5K | 1.0M | 1.2M | 10.7M | 325.8K | 11.0M | 10.7M |
| # leaves | 419 | 26.0K | 62 | 35.6K | 4K | 88 | 4.2K | 4.0K |
| avg degree | 88.28 | 6.28 | 175.78 | 15.66 | 460.92 | 4.08 | 288.04 | 460.92 |
| avg distance | 2.79 | 4.29 | 2.31 | 4.89 | 2.45 | 115.62 | 3.55 | 2.45 |
| global clustering | 0.23 | 0.02 | 0.23 | 0.26 | 0.27 | 0.01 | 0.24 | 0.27 |
| avg local clustering | 0.53 | 0.03 | 0.66 | 0.41 | 0.70 | 0.00 | 0.85 | 0.70 |
| degree assortativity | $-0.08$ | $-0.15$ | $-0.11$ | 0.01 | $-0.35$ | 0.66 | $-0.30$ | $-0.35$ |
| # classes | 2 | 2 | 2 | 2 | 21 | — | — | — |
| adjusted homophily | 0.09 | 0.02 | 0.04 | 0.59 | 0.08 | — | — | — |
| label informativeness | 0.01 | 0.00 | 0.01 | 0.31 | 0.02 | — | — | — |
| target assortativity | — | — | — | — | — | 0.66 | 0.19 | 0.13 |
| % labeled nodes | 100 | 100 | 72 | 93 | 100 | 23 | 100 | 100 |
| # num features | 6 | 19 | 22 | 14 | 1 | 2 | 4 | 0 |
| # bin features | 2 | 11 | 1 | 0 | 0 | 16 | 0 | 0 |
| # cat features | 1 | 1 | 1 | 0 | 6 | 5 | 13 | 11 |

**city-roads**  This dataset is obtained from the logs of a navigation service and represents the road network of a major city. In this dataset, city roads are considered as graph nodes, and there is a directed edge from a node $i$ to another node $j$, if the corresponding roads are incident to the same crossing, and moving from $i$ to $j$ is permitted by traffic rules. Thus, the obtained graph is directed, and we extract its largest weakly connected component. The set of features includes such attributes as the length of roads, the number of segments they consist of, the speed limits, various categories related to the properties of road surface, and numerous indicators describing the permission for different types of vehicles. The task is to predict the travel speed on roads at a specific timestamp.

**avazu-devices**  This is another open-source dataset that has been introduced at the Kaggle competition organized by Avazu (Wang & Cukierski, 2014). It represents the interactions between devices and advertisements on the internet. This dataset is originally bipartite and contains entities of three types — devices, sites that are visited by these devices, and applications that are used to visit them. A version of this dataset has been used by Ivanov & Prokhorenkova (2021) in their study. However, it contained only a small subset of interactions from the original dataset, which resulted in a small-sized graph. Because of that, we decided to consider the whole dataset and transform it by projecting to the part of devices. Let $\mathbf{P} \in \mathbb{R}^{m \times n}$ be a weighted adjacency matrix of devices and entry points defined as the combinations of sites and applications, where $m$ is the number of entry points, $n$ is the number of devices, and $p_{ij}$ denotes how many times device $j$ has used entry point $i$ (i.e., visited a specific site under a specific application). Then, $\mathbf{B} = \mathbf{P}^T \mathbf{P} \in \mathbb{R}^{n \times n}$ corresponds to the weighted adjacency matrix of devices, where $b_{ij}$ is the dot product of columns $i$ and $j$ in $\mathbf{P}$. The interpretation of this matrix is similar to what we discussed above for hm-products. Finally, we obtain a binary adjacency matrix $\mathbf{A} \in \{0, 1\}^{n \times n}$ of devices with $a_{ij} = [b_{ij} \geqslant \gamma]$ by applying a threshold $\gamma = 1000$ to the weights $b_{ij}$. The resulting network is undirected. The set of features includes device model and type, banner positions, and numerous categorical features that have been already anonymized before being released to public access. The task is to predict the click-through rate (CTR) observed on devices.

## 3.2 Diversity of the benchmark

A key property of our benchmark is its diversity. As described above, our graphs come from different domains and have different prediction tasks. Their edges are also constructed in different ways (based on user interactions, activity similarity, etc.). However, the proposed datasets also differ in many other ways. Some properties of our graphs are presented in Table 1. First, note that the sizes of our graphs range from 11K to 170K nodes. The smaller graphs can be suitable for compute-intensive models, while the larger graphs can provide a scaling challenge. The average degree of our graphs also varies significantly — most graphs have the average degree of tens or hundreds, which is much larger than for most datasets used in graph machine learning research; however, we also have some sparser graphs such as `questions-tab` and `city-roads`. The average distance between two nodes in our graphs differs from 2.31 in `amazon-users` to 115.62 in `city-roads`. Further, we report the values of clustering coefficients which show how typical closed node triplets are for the graph. In the literature, there are two definitions of clustering coefficients (Boccaletti et al., 2014): global and average local ones. We have graphs where both clustering coefficients are high or almost zero. The degree assortativity coefficient is defined as the Pearson correlation coefficient of degrees among pairs of linked nodes. Most of our graphs have negative degree assortativity, which means that nodes tend to connect to other nodes with dissimilar degrees, while for the `city-roads` dataset the degree assortativity is positive and large.

Further, let us discuss the graph-label relations in our datasets. A classification dataset is considered homophilous, if nodes tend to connect to nodes of the same class, and it is considered heterophilous if nodes tend to connect to nodes of different classes. We use adjusted homophily to characterize homophily level, as it has been shown to have more desirable properties than other homophily measures used in the literature (Platonov et al., 2022). We can see that `city-reviews` dataset is homophilous, while the rest of our classification datasets are heterophilous. One more characteristic to describe graph-label relationships is label informativeness (Platonov et al., 2022). It shows how much information about the label of a given node can be derived from the labels of neighbor nodes. In our dataset, label informativeness correlates with adjusted homophily, which is typical for labeled graphs. To measure the similarity of labels for regression datasets, we use the target assortativity coefficient — the Pearson correlation coefficient of target values between pairs of connected nodes. For instance, on the `city-roads` dataset, the target assortativity is positive, while for the other two regression datasets it is negative.

Note that some of our graphs contain unlabeled nodes. This is a typical situation for industry and science, yet it is underrepresented in graph machine learning benchmarks. Unlabeled nodes give an additional advantage to graph-aware models, as they can utilize the information about the features of these nodes and their position in the graph even without knowing their labels.

Finally, our datasets have tabular features with different numbers and balance of numerical, binary, and categorical attributes. The numerical features have widely different scales and distributions. For example, in `questions-tab` dataset, most of the features are counters (subscribers count, achievements count, articles count) which have Poisson-like distributions, while the rating feature has a very different distribution with possible negative values and lots of outliers.

Overall, our datasets are diverse in domain, scale, structural properties, graph-label relations, and node attributes. Providing meaningful prediction tasks, they may serve as a valuable tool for the research and development of machine learning models for processing graph-structured data with heterogeneous features.

## 4 Baseline models

Here, we briefly discuss the machine learning methods for working with tabular features and graph-structured data that we evaluate in our experiments.

**A simple baseline** As a simple baseline, we use a ResNet-like model (He et al., 2016). It does not have any information about the graph structure and operates on nodes as independent samples (we call such models *graph-agnostic*). This model also does not have any specific designs for working with tabular features.

**Classical tabular models** We consider three most popular implementations of GBDT: XGBoost (Chen & Guestrin, 2016), LightGBM (Ke et al., 2017), and CatBoost (Prokhorenkova et al., 2018). These models are also graph-agnostic.

**Tabular deep learning models** We use two graph-agnostic deep learning models that have been recently proposed for working with tabular data. First, we use the numerical feature embedding technique introduced by Gorishniy et al. (2022). This technique adds a learnable module that transforms numerical features in such a way that they can be better processed by neural networks. In particular, we use a simple MLP with PLR (`Periodic-Linear-ReLU`) numeric feature embeddings as this combination has shown the best performance in the original study. We refer to this model as MLP-PLR. Further, we consider retrieval-augmented models for tabular data and use the recently proposed TabR model (Gorishniy et al., 2023) as it has shown strong performance on a wide range of tabular datasets. Note that TabR also uses PLR numerical feature embeddings.

**Graph machine learning models** We consider several representative GNN architectures for our experiments. First, we use GCN (Kipf & Welling, 2016) and GraphSAGE (Hamilton et al., 2017) as simple classical GNN models. Further, we use two attention-augmented GNNs: GAT (Veličković et al., 2017) and Graph Transformer (GT) (Shi et al., 2020), along with their simple modifications GAT-sep and GT-sep that add ego-embedding and neighborhood-aggregated-embedding separation to the models, as proposed in Platonov et al. (2023). We equip all GNNs with skip-connections (He et al., 2016) and layer normalization (Ba et al., 2016), which we found very important for the stability and strong performance. Our GNNs are implemented in the same codebase as ResNet — we simply swap each residual block of ResNet with the residual neighborhood aggregation block of the selected GNN architecture. Thus, comparing the performance of ResNet and GNNs is the most direct way to see if graph information is helpful for the task.

**Specialized models** We also use two recently proposed methods for solving node property prediction problems on graphs with heterogeneous features: BGNN (Ivanov & Prokhorenkova, 2021) and EBBS (Chen et al., 2022). Unfortunately, the official implementation of EBBS does not work for node classification datasets, thus we only evaluate this method on node regression tasks.

## 5 EXPERIMENTS

**Setup** In our experiments, the labeled nodes are split into train, validation, and test parts in proportion $50 : 25 : 25$. We use stratified split for classification datasets and random split for regression datasets. We report ROC-AUC for binary classification, Accuracy for multi-class classification, and $R^2$ (coefficient of determination) for regression. We run each model 5 times and report the mean and standard deviaton of each metric value.

For GBDT and tabular deep learning models, we conduct an extensive hyperparameter search using Optuna (Akiba et al., 2019) (see Appendix D for the details about the search space), whereas for specialized model we follow the recommendations from the original studies. In particular, we choose a default set of hyperparameters for EBBS in regression task, as the authors claim that they should work well across different tasks and graph datasets; for BGNN, we use the experimental pipeline proposed in the official implementation of this method, which performs hyperparameter tuning over a pre-defined grid of values.

For GNNs, we found that, when augmented with skip-connections and layer normalization, they are not very sensitive to the hyperparameter choice, thus we did not tune them for graph neural models. The set of hyperparameter used in our experiments is described in Appendix C.

To conduct experiments with GBDT and tabular deep learning models, we use the source code from TabR repository (Gorishniy et al., 2023), while for the two specialized models BGNN and EBBS, we use their corresponding repositories (Ivanov & Prokhorenkova, 2021; Chen et al., 2022).

When using deep learning models for tabular data, the pre-processing of numerical features is critically important. In our preliminary experiments, we found that quantile transformation to normal distribution typically works best, thus we used it in all our experiments. For categorical features, we used one-hot encoding for all models, except for CatBoost, which is designed specifically to handle such features.

Table 2: Classification results. Accuracy is reported for `hm-groups` dataset, ROC-AUC is reported for the remaining datasets. Best results are marked with cyan.

|  | tolokers-tab | questions-tab | amazon-users | city-reviews | hm-groups |
|---|---|---|---|---|---|
| ResNet | $75.85 \pm 0.19$ | $84.80 \pm 0.30$ | $89.59 \pm 0.56$ | $89.88 \pm 0.02$ | $70.71 \pm 0.20$ |
| XGBoost | $78.59 \pm 0.30$ | $87.19 \pm 0.12$ | $90.93 \pm 0.15$ | $93.04 \pm 0.02$ | $71.08 \pm 0.70$ |
| LightGBM | $78.60 \pm 0.08$ | $87.33 \pm 0.07$ | $90.86 \pm 0.14$ | $93.01 \pm 0.01$ | $71.08 \pm 0.08$ |
| CatBoost | $77.98 \pm 0.13$ | $87.76 \pm 0.04$ | $91.26 \pm 0.56$ | $93.03 \pm 0.01$ | $71.14 \pm 0.12$ |
| MLP-PLR | $77.24 \pm 0.43$ | $87.69 \pm 0.21$ | $91.18 \pm 0.54$ | $92.30 \pm 0.05$ | $71.04 \pm 0.14$ |
| TabR | $77.62 \pm 0.86$ | $86.93 \pm 0.33$ | $90.95 \pm 0.09$ | $92.34 \pm 0.13$ | $71.44 \pm 0.29$ |
| GCN | $85.54 \pm 0.20$ | $82.83 \pm 2.34$ | $90.19 \pm 0.26$ | $91.86 \pm 0.12$ | $76.40 \pm 0.08$ |
| GraphSAGE | $83.91 \pm 0.07$ | $88.25 \pm 0.42$ | $90.16 \pm 0.26$ | $92.64 \pm 0.05$ | $81.01 \pm 0.29$ |
| GAT | $84.78 \pm 0.12$ | $87.07 \pm 0.46$ | $90.46 \pm 0.41$ | $92.24 \pm 0.06$ | $82.82 \pm 0.31$ |
| GAT-sep | $85.03 \pm 0.37$ | $88.41 \pm 0.35$ | $90.23 \pm 0.42$ | $92.56 \pm 0.07$ | $85.62 \pm 0.44$ |
| GT | $82.99 \pm 0.29$ | $86.38 \pm 0.34$ | $90.11 \pm 0.32$ | $92.33 \pm 0.05$ | $86.84 \pm 0.52$ |
| GT-sep | $83.04 \pm 0.22$ | $86.94 \pm 0.48$ | $90.34 \pm 0.21$ | $92.18 \pm 0.03$ | $85.68 \pm 0.44$ |
| GCN-PLR | $85.63 \pm 0.13$ | $89.52 \pm 0.43$ | $91.28 \pm 0.15$ | $92.59 \pm 0.07$ | $76.68 \pm 0.21$ |
| GraphSAGE-PLR | $84.81 \pm 0.08$ | $90.26 \pm 0.28$ | $91.65 \pm 0.05$ | $93.36 \pm 0.10$ | $80.95 \pm 0.22$ |
| GAT-PLR | $85.67 \pm 0.22$ | $89.54 \pm 0.16$ | $90.78 \pm 0.12$ | $92.56 \pm 0.12$ | $81.70 \pm 0.36$ |
| GAT-sep-PLR | $85.85 \pm 0.10$ | $90.25 \pm 0.30$ | $91.68 \pm 0.08$ | $93.17 \pm 0.22$ | $85.28 \pm 0.45$ |
| GT-PLR | $83.87 \pm 0.54$ | $88.30 \pm 0.80$ | $90.16 \pm 0.22$ | $92.78 \pm 0.17$ | $85.83 \pm 0.37$ |
| GT-sep-PLR | $84.19 \pm 0.22$ | $89.63 \pm 0.29$ | $91.82 \pm 0.17$ | $93.09 \pm 0.16$ | $85.74 \pm 0.08$ |
| BGNN | $78.37 \pm 0.55$ | $70.87 \pm 0.87$ | $83.66 \pm 0.38$ | $90.35 \pm 0.13$ | $84.60 \pm 0.50$ |

**Results** In this subsection, we analyze the results of our experiments. The results for classification and regression datasets are provided in Tables 2 and 3, respectively.

First, we note that for all datasets vanilla ResNet achieves worse results than the best of GBDT and tabular deep learning models. This shows that our datasets indeed have meaningful tabular features, and methods designed specifically for tabular data outperform vanilla deep learning approaches.

Second, we can see that the considered GNNs outperform ResNet on almost all datasets, with the exception of GCN on `questions-tab` and `city-roads`. Since our ResNet and GNNs are implemented in the same codebase and thus directly comparable, this shows that the graph structure in our datasets is helpful for the given tasks. Compared to the best graph-agnostic models, the best of vanilla GNNs outperform them on all datasets except for `amazon-users` and `city-reviews`. If we additionally consider PLR-augmented GNNs, then the best of GNNs always outperform graph-agnostic models.

However, we note that the increase in performance achieved by GNNs compared to graph-agnostic models may vary significantly across different datasets. For example, on `hm-groups` dataset, the best graph-agnostic model achieves 71.44 points of Accuracy, while the best vanilla GNN achieves 86.84 points, which is a very large difference. In contrast, on `city-reviews` dataset, the best graph-agnostic model achieves 93.04 ROC-AUC, the best vanilla GNN achieves 92.64 ROC-AUC, and the best PLR-augmented GNN achieves 93.36 ROC-AUC. However, we note that for industrial applications such as fraud detection, even such small improvements can be critical.

The relative performance of GNNs also varies between datasets, and the best GNN architecture may differ from dataset to dataset. For instance, GCN is the best on `tolokers-tab`, GraphSAGE is the best on `city-roads`, GAT is the best on `hm-prices`, and GT is the best on `questions-tab`. Similarly, the -sep modification sometimes improves GNN performance significantly and sometimes does not. Thus, it is hard to make a prior choice of architecture, and comparative experiments are required for each dataset.

Further, we can see that the recently proposed models designed specifically for graphs with tabular features, namely BGNN and EBBS, failed to produce results competitive with vanilla GNNs on our datasets, with the exception of BGNN on `hm-groups` and `city-roads` datasets.

Finally, let us note that PLR embeddings for numerical features often improve the performance of GNNs, and sometimes this improvement is very significant (for example, on `questions-tab` dataset, GCN achieves 82.83 ROC-AUC, while GCN-PLR achieves 89.52 ROC-AUC). Overall, PLR-augmented GNNs provide the best results on 6 out of 8 datasets.

Table 3: Regression results. $R^2$ is reported for all datasets. OOM denotes the only experiment that could not be carried out due to the GPU memory constraints. Best results are marked with cyan.

|  | city-roads | avazu-devices | hm-prices |
|---|---|---|---|
| ResNet | $51.85 \pm 0.40$ | $20.33 \pm 0.29$ | $58.74 \pm 0.83$ |
| XGBoost | $57.07 \pm 0.02$ | $25.07 \pm 0.03$ | $67.63 \pm 0.08$ |
| LightGBM | $56.35 \pm 0.06$ | $24.69 \pm 0.04$ | $67.76 \pm 0.08$ |
| CatBoost | $56.68 \pm 0.12$ | $26.01 \pm 0.18$ | $68.40 \pm 0.28$ |
| MLP-PLR | $56.88 \pm 0.09$ | $23.08 \pm 0.36$ | $69.05 \pm 0.05$ |
| TabR | $56.44 \pm 0.07$ | OOM | $67.94 \pm 0.54$ |
| GCN | $50.86 \pm 1.07$ | $20.99 \pm 0.19$ | $68.09 \pm 0.45$ |
| GraphSAGE | $57.79 \pm 0.55$ | $26.58 \pm 0.29$ | $66.65 \pm 0.35$ |
| GAT | $53.44 \pm 0.75$ | $24.92 \pm 0.44$ | $72.69 \pm 0.61$ |
| GAT-sep | $57.25 \pm 0.44$ | $25.49 \pm 0.14$ | $68.16 \pm 0.89$ |
| GT | $55.92 \pm 0.35$ | $27.07 \pm 0.17$ | $64.06 \pm 1.17$ |
| GT-sep | $58.49 \pm 0.38$ | $25.46 \pm 0.74$ | $68.76 \pm 1.12$ |
| GCN-PLR | $48.02 \pm 0.91$ | $20.82 \pm 0.55$ | $67.85 \pm 0.56$ |
| GraphSAGE-PLR | $57.27 \pm 0.90$ | $26.58 \pm 0.29$ | $66.64 \pm 0.72$ |
| GAT-PLR | $54.94 \pm 0.86$ | $24.88 \pm 0.31$ | $73.02 \pm 0.62$ |
| GAT-sep-PLR | $57.57 \pm 0.60$ | $24.93 \pm 0.29$ | $69.95 \pm 1.69$ |
| GT-PLR | $57.75 \pm 0.37$ | $25.19 \pm 0.65$ | $63.86 \pm 1.55$ |
| GT-sep-PLR | $59.49 \pm 0.37$ | $23.39 \pm 0.30$ | $69.70 \pm 1.72$ |
| BGNN | $57.87 \pm 1.73$ | $21.81 \pm 0.33$ | $62.37 \pm 9.66$ |
| EBBS | $25.28 \pm 1.99$ | $11.94 \pm 0.04$ | $30.49 \pm 2.83$ |

Based on these results, we make the following recommendations for researchers and practitioners working with tabular data:

- If it is possible to define meaningful relations between data samples, it is worth trying to convert the data to a graph and experimenting with graph machine learning methods — it can lead to significant gains.

- The best GNN architecture for graphs with tabular features strongly depends on the specific dataset, and it is important to try different architectures.

- Standard GNNs generally perform better than the recently proposed models designed specifically for graphs with tabular features.

- PLR embeddings for numerical features are a simple modification that can significantly improve the performance of GNNs on graphs with tabular features, and it is worth trying to incorporate them in graph neural architecture.

## 6 CONCLUSION

In this work, we introduce TabGraphs — the first benchmark of graph datasets with heterogeneous tabular node features. Our datasets are diverse and have realistic prediction tasks. We have evaluated a large number of methods from tabular machine learning and graph machine learning on this benchmark, including models that combine techniques from both fields. Based on our experimental findings, we provide recommendations and insights for researchers and practitioners working with tabular and graph data. In particular, we show that transforming tabular data relations to graph structure and using graph machine learning methods can significantly increase predictive performance in many real-world tasks. We believe that our benchmark and insights obtained using it will be helpful for both industry and science and will facilitate further research on the intersection of tabular and graph machine learning.

## 7 REPRODUCIBILITY STATEMENT

In Section 3, we describe the proposed graph datasets, including their original sources and methodologies employed to prepare them for our study. In Section 5, we also provide the details about our experimental setup, specifying the references to the open-source implementations of the used baselines. The configurations of machine learning models are also reported in Appendices D and C. To ensure reproducibility of our work, we plan to release both the source code and the datasets once we finalize the required legal procedures.

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

# A    ON GRAPH CONSTRUCTION IN TABGRAPHS

In this section, we discuss some important aspects of constructing graphs in our benchmark.

For all of the proposed graph datasets, we use external relational information for constructing the underlying graph. Another possible approach would be deriving the connections between data samples from their node features, but it is not trivial in the case of heterogeneous tabular features, as these features have widely different scales, distributions, meaning and importance for the task. However, comparing their representation from some layer of a neural network is more meaningful. This is exactly what retrieval-augmented models do (in essence, they simultaneously learn sample representations and a graph based on their similarity). We discuss these models in the related work section and include TabR, a representative example of them, in our experiments.

Further, some of the proposed graph datasets are derived from the corresponding bipartite networks of interactions between entities of different types, which is done by projecting the original network to one of its parts. This holds for `city-reviews`, which connects users that write their reviews for the same organizations, `hm-products`, where products are connected if they are bought by the same customers, and `avazu-devices`, for which we connect devices that use the same entry point. As described in Section 3, for this purpose, we use an additional parameter $\gamma$, which depends on a particular graph dataset and serves as a threshold to control the density of the underlying graph. To construct our benchmark, we adjust this parameter $\gamma$ so that the resulting graphs are not too sparse and not too dense. Although the mentioned datasets can be constructed in many different ways depending on the threshold $\gamma$, we assume that our approach is quite reasonable, since even with such parameter choice we manage to achieve higher performance on all these datasets by using GNN models that exploit graph information and thus demonstrate the benefits of knowing graph structure in the considered problems.

Finally, preparing the graph might take some time. The exact amount varies depending on the nature of the raw data and the type of edges used in the graph being constructed. For example, if the edges are based on interactions between data samples (e.g., in `questions-tab`), then constructing the graph is relatively fast, as we simply need to obtain a list of interactions, and each interaction corresponds to one edge. On the contrary, if the edges are based on some kind of activity similarity (e.g., in `avazu-devices`), constructing the edges might take more time, as we need to assess how similar two data samples are in terms of their activity. However, the construction of a graph only needs to be performed once, as part of data pre-processing, so the time required for it is negligible compared to the time spent for running multiple experiments on the dataset.

# B    PREVIOUS GRAPH DATASETS WITH TABULAR FEATURES

In this section, we discuss the issues of some graph datasets with tabular features used for evaluation in previous studies (Ivanov & Prokhorenkova, 2021; Chen et al., 2022).

Considering the classification datasets, `dblp` and `slap` actually have homogeneous features, which were augmented with some graph statistics to make them heterogeneous. Further, the corresponding graphs are actually heterogeneous information networks (HINs), which have several relation types, yet only one relation type was used. As for the `house-class` and `vk-class` datasets, these are actually regression datasets that were converted to classification datasets by binning target values.

Regarding the regression datasets, `county` and `avazu` are very small. In our benchmark, we adapt an extended version of the `avazu` dataset, which has a significantly different scale (please refer to Table 1). Further, the `house` dataset originally did not contain a graph, and edges were constructed based on physical proximity, while the original features representing geographic coordinates were removed. However, keeping these features and removing the graph leads to similar performance, which shows that the graph is not necessary for this task. Finally, we found that for the `vk` dataset, CatBoost, GCN, and GAT achieve the $R^2$ values less than $0.1$, which shows that the provided features and graph structure are not very helpful for the task.

## C    HYPERPARAMETERS OF GRAPH NEURAL NETWORKS

In our experiments, we found that, when augmented with skip-connections and layer normalization, our GNNs are not very sensitive to the choice of hyperparameters. Thus, we used a single set of hyperparameters, which is described below.

The number of GNN blocks is 3. The hidden dimension is $512$. The dropout probability is $0.2$. For GAT and GT models, the number of attention heads is set to $4$. We use the GELU activation function (Hendrycks & Gimpel, 2016) in all our GNNs. We also use the Adam optimizer (Kingma & Ba, 2014) with learning rate of $3 \cdot 10^{-5}$. We train each model for $1000$ steps and select the best step based on the performance on the validation subset. Our baselines are implemented using PyTorch (Paszke et al., 2019) and DGL (Wang et al., 2019).

PLR embeddings for numerical features have additional hyperparameters, and, as noted in (Gorishniy et al., 2022), tuning them can be very important. However, for a fair comparison with vanilla GNNs, we use these embeddings without tuning. The hyperparameters are the following: number of frequencies is $48$, frequency scale is $0.01$, embedding dimension is $16$. We use the "lite" version of PLR introduced in (Gorishniy et al., 2023).

## D    HYPERPARAMETERS OF CLASSIC AND DEEP LEARNING MODELS

In Tables 4-8, we provide the hyperparameter search space for classic and deep learning models. In our experiments, we perform 70 rounds of Bayesian optimization using Optuna (Akiba et al., 2019).

Table 4: The hyperparameter tuning space for XGBoost.

| Parameter | Distribution |
|---|---|
| colsample_bytree | $\text{Uniform}[0.5, 1.0]$ |
| gamma | $\{0.0, \text{LogUniform}[0.001, 100.0]\}$ |
| lambda | $\{0.0, \text{LogUniform}[0.1, 10.0]\}$ |
| learning_rate | $\text{LogUniform}[0.001, 1.0]$ |
| max_depth | $\text{UniformInt}[3, 14]$ |
| min_child_weight | $\text{LogUniform}[0.0001, 100.0]$ |
| subsample | $\text{Uniform}[0.5, 1.0$ |

Table 5: The hyperparameter tuning space for LightGBM.

| Parameter | Distribution |
|---|---|
| feature_fraction | $\text{Uniform}[0.5, 1.0]$ |
| lambda_l2 | $\{0.0, \text{LogUniform}[0.1, 10.0]\}$ |
| learning_rate | $\text{LogUniform}[0.001, 1.0]$ |
| num_leaves | $\text{UniformInt}[4, 768]$ |
| min_sum_hessian_in_leaf | $\text{LogUniform}[0.0001, 100.0]$ |
| bagging_fraction | $\text{Uniform}[0.5, 1.0]$ |

Table 6: The hyperparameter tuning space for CatBoost.

| Parameter | Distribution |
|---|---|
| bagging_temperature | Uniform[0.0, 1.0] |
| depth | UniformInt[3, 14] |
| l2_leaf_reg | Uniform[0.1, 10.0] |
| leaf_estimation_iterations | Uniform[1, 10] |
| learning_rate | LogUniform[0.001, 1.0] |

Table 7: The hyperparameter tuning space for MLP.

| Parameter | Distribution |
|---|---|
| num_layers | UniformInt[1, 6] |
| hidden_size | UniformInt[64, 1024] |
| dropout_rate | {0.0, Uniform[0.0, 0.5]} |
| learning_rate | LogUniform[3$e$-5, 1$e$-3] |
| weight_decay | {0, LogUniform[1$e$-6, 1$e$-3]} |
| plr_num_frequencies | UniformInt[16, 96] |
| plr_frequency_scale | LogUniform[0.001, 100.0] |
| plr_embedding_size | UniformInt[16, 64] |

Table 8: The hyperparameter tuning space for TabR.

| Parameter | Distribution |
|---|---|
| num_encoder_blocks | UniformInt[0, 1] |
| num_predictor_blocks | UniformInt[1, 2] |
| hidden_size | UniformInt[96, 384] |
| context_dropout | Uniform[0.0, 0.6] |
| dropout_rate | Uniform[0.0, 0.5] |
| learning_rate | LogUniform[3$e$-5, 1$e$-3] |
| weight_decay | {0, LogUniform[1$e$-6, 1$e$-3]} |
| plr_num_frequencies | UniformInt[16, 96] |
| plr_frequency_scale | LogUniform[0.001, 100.0] |
| plr_embedding_size | UniformInt[16, 64] |

