# OpenReview forum: "TabGraphs: new benchmark and insights for learning on graphs with tabular features"
_ICLR.cc/2024/Conference — Submitted to ICLR 2024_

### Official Review · Reviewer_b5gj · 2023-10-29

**Soundness:** 3 good
**Presentation:** 3 good
**Contribution:** 3 good
**Rating:** 6
**Confidence:** 3

**Summary:**

The paper says that tabular data can be seen as graph-structured data to improve the predictive performance,  which is interesting. The paper proposes datasets with heterogenous node features for future research that fills in the gap between machine learning for tabular data and graph machine learning, which makes sense. Most of the proposed datasets are processed by certain rules, which undermines the credibility of the datasets and insights.

**Strengths:**

S1. Filling in the gap between machine learning for tabular  data and graph machine learning make senses.

S2. Datasets of different kinds are included. Major models are evaluated.

S3. The paper reads well.

**Weaknesses:**

W1. It is unclear that how much model performance can be achieved from the proposed heterogeneous features. See Q1.

W2. Important data is missing. Strings or texts are common data types in Tabular data. TabGraph does not include strings or texts data (see Q2). Data from science domain is not included (see Q3).

W3. Insights are not completely convincing as datasets are preprocessed by rules. See Q4 and Q5.

W4. Limitations of the paper are not discussed in the paper. Please discuss limitations of the paper.

**Questions:**

Q1: Model performances on homogeneous features are not shown. Can authors compare models performances on hemogeneous features and heterogeneous features to show the efficacy of  proposed datasets?

Q2:  Can authors explain why strings or texts data are not included by TabGraph?

Q3: Graph data from science domain is not included. Can authors add one dataset for science for node classification and node regression tasks?

Q4: The paper says that " In our preliminary experiments, we found that by using GBDT on all the features available in this dataset, one may achieve ∼ 0.99 ROC-AUC, so we decided to remove the number of votes as the most important feature .." If GBDT performs very well on certain types of data, insights probably need to be updated.  Can authors comment on why the number of votes shoud be removed?

Q5: Can authors explain why certain thresholds are chosen for different datasets in Section 3.1?

Q6: Table 1 shows that there are three types of features: number features,  binary features, category features. Can authors show which feature is more important than others?

Q7: How are the node features extended to be heterogeneous for tolokers-tab dataset?

Q8: Can authors provide tentative dates to release both the source code and the datasets?

Q9: Can authors explain how expensive is transforming tabular data relations to graph structure?

---

> ### Author Response · Authors · 2023-11-16
>
> Thank you for the review and detailed feedback! We answer to the questions below.
>
> Q1. We would like to point out that for different kinds of datasets different types of features (e.g., homogeneous or heterogeneous) are available. In our work, we specifically focus on heterogeneous features not because they are meant to provide better or worse performance, but because for many real-world datasets this is the only kind of features available, and thus designing models specifically for working with such features is important. For this reason, a good benchmark of datasets with such features is required.
>
> Q2. Before feeding into an ML model, strings of text are typically preprocessed using bag-of-words and other text embedding models. These algorithms produce homogeneous features. As we have discussed in the related work section, there is already a vast number of popular datasets with such text-based homogeneous features. In our paper, we specifically focus on graphs with heterogeneous features, which can often be found in real-world tasks, but for which good datasets are currently lacking.
>
> Note that tabular datasets that do not have any strings of text as features are quite widespread in the real world. For example, among the datasets considered in our paper, only questions-tab had strings of text that could be used as additional features. These features were used as the only feature type in the original version of the questions dataset introduced in [1]. However, we removed them in our version of the dataset, as we have found out that, in this particular case, models perform much better with heterogeneous features, which can be seen by comparing the results of GNNs in our paper and [1] on this dataset.
>
> Q3. Indeed, it would be very useful to introduce more datasets from different sciences, but, unfortunately, we do not have access to such datasets. We will continue looking for them, and if you are aware of any quality open scientific datasets with heterogeneous features and additional graph structure, we will be glad to include them in our benchmark.
>
> Q4. In fact, all of the considered models can achieve almost perfect metrics on this dataset with the full feature set (we will clarify this in the paper). This dataset is based on the data from [2]. From the description of the data, we hypothesize that the number of votes was used for constructing the target variable, and thus introduces a data leak. Because of that, we decided to remove this feature.
>
> Q5. The thresholds were chosen so as to produce a graph that is not too sparse and not too dense. Indeed, this threshold can be viewed as a hyper-parameter of graph construction, however, in our paper, we show that the produced graphs are helpful for the task even without searching over the values of this parameter.
>
> Q6. This part of the table is meant to show that such a mix of different types of features that is typical for tabular datasets is indeed present in our datasets. However, even between the features of one type there can be a drastic difference in their importance.
>
> Q7. In the original tolokers dataset from [1], features are processed to make them easy for GNN models to operate on. For our work, we have obtained the original set of features and used it without preprocessing beyond scaling, in line with how it is typically done in tabular machine learning. The features are various performance statistics of workers, such as the number of approved and skipped assignments (numerical features), as well as worker profile information such as their education level (categorical feature).
>
> Q8. We can release the datasets that do not contain proprietary data (tolokers-tab, questions-tab, amazon-users) now. However, for other datasets we need to finalize the legal procedures, which we will do upon the acceptance of the paper.
>
> Q9. Indeed, preparing the graph might take some time. The exact amount varies depending on the nature of the raw data and the type of edges used in the graph being constructed.
>
> For example, if the edges are based on interactions between data samples (e.g., in questions-tab), then constructing the graph is relatively fast, as we simply need to obtain a list of interactions and each interaction corresponds to one edge. On the contrary, if the edges are based on some kind of activity similarity (e.g., in avazu-devices), constructing the edges might take more time as we need to assess how similar two data samples are in terms of their activity.
>
> However, the construction of a graph only needs to be performed once, as part of data preprocessing, so the time required for it is negligible compared to the time spent for running multiple experiments on the dataset. We will add a discussion of this point to the paper.
>
> [1] “A critical look at the evaluation of GNNs under heterophily: are we really making progress?” in ICLR, 2023
>
> [2] “GCN-Based User Representation Learning for Unifying Robust Recommendation and Fraudster Detection” in SIGIR, 2020

---

> > ### Comment · Reviewer_b5gj · 2023-11-20
> >
> > I appreciate the replies from authors.  Comments and input from other reviewers make sense to me.

---

### Official Review · Reviewer_VTYd · 2023-11-01

**Soundness:** 2 fair
**Presentation:** 3 good
**Contribution:** 2 fair
**Rating:** 3
**Confidence:** 4

**Summary:**

This paper proposed a benchmark (TabGraph) for learning on graphs with heterogeneous tabular node features. The benchmark contains 8 datasets that vary in the mixture of feature types, node numbers, average degrees and domain. The author compared GNN models, tabular deep learning models, tabular tree models, and some tabular and graph hybrid models. The author also proposed to embed the tabular node features with the PLR algorithm and demonstrated that it can boost the performance.

**Strengths:**

1. Lots of tabular data have correlated samples that can be effectively modeled by a graph. The TabGraphs benchmark compared several recent methods under this setting, and is helpful for people dealing with such kind of tabular data.
2. The author showed that PLR can boost the performance of GNN on graphs with heterogeneous tabular features.

**Weaknesses:**

1. Although PLR can boost the performance of GNN, it is not novel. It is not surprising that a better node representation can boost the performance of GNN.
2. Although the author claimed that these 8 datasets in TabGraph is diverse, their selection criteria is unclear to the reader. In fact, lots of tabular benchmarks contain hundreds of datasets (e.g., AMLB). Benchmarking on only 8 tables may not be conclusive.

**Questions:**

What's the selection criteria of these 8 tabular datasets?
For generic tabular datasets, you can build a graph that connects the samples (e.g., based on the sample correlation). Can GNN + PLR still obtain performance in the general setting?

---

> ### Author Response · Authors · 2023-11-16
>
> Thank you for the review and questions! We address your concerns below.
>
> > Although PLR can boost the performance of GNN, it is not novel. It is not surprising that a better node representation can boost the performance of GNN.
>
> Indeed PLR is not novel (and we do not claim it to be). However, it was never used in the graph ML setting before, and our work highlights that it can be an easy way to improve performance on realistic graph datasets with heterogeneous tabular features.
>
> > Although the author claimed that these 8 datasets in TabGraph is diverse, their selection criteria is unclear to the reader. In fact, lots of tabular benchmarks contain hundreds of datasets (e.g., AMLB). Benchmarking on only 8 tables may not be conclusive.
>
> For our benchmark, we have specifically selected datasets with available external information about the relationship between data samples that can be used for constructing a graph. Such datasets are much more difficult to find than standard tabular datasets. If you are aware of other realistic tabular datasets with such external information, please let us know so that we could include them in our benchmark.
>
> Note that, in our work, we consider different kinds of relations between data samples — they are based on interactions (e.g., in the questions-tab dataset), activity similarity (e.g., in the avazu-devices dataset), physical connection (e.g., in the city-roads dataset). Importantly, we show that all these relation types can provide useful information to the models that can leverage graph-structured data.
>
> > What's the selection criteria of these 8 tabular datasets? For generic tabular datasets, you can build a graph that connects the samples (e.g., based on the sample correlation). Can GNN + PLR still obtain performance in the general setting?
>
> As mentioned above, our aim was to select datasets that have meaningful prediction tasks and contain external relational information, while being quite diverse in terms of tabular feature distribution and underlying graph structure.
>
> Note that directly comparing samples based on their features is not trivial in the case of heterogeneous tabular features (on which we focus in our paper), as these features have widely different scales, distributions, meaning and importance for the task. However, comparing their representation from some layer of a neural network is more meaningful. This is exactly what retrieval-augmented models do (in essence, they simultaneously learn sample representations and a graph based on their similarity). We discuss these models in the related work section and include TabR, a representative example of them, in our experiments. If you know some other meaningful ways to measure the similarity between objects with heterogeneous tabular features for constructing relations between them, please let us know so that we could extend our empirical study.

---

### Official Review · Reviewer_Umnm · 2023-11-01

**Soundness:** 2 fair
**Presentation:** 2 fair
**Contribution:** 2 fair
**Rating:** 3
**Confidence:** 4

**Summary:**

The paper addresses the lack of ml models for graphs with heterogeneous features. The work provides several benchmark datatsets with such characteristics and evaluates with numerous methods to analyze the performances. Finally, the paper provides some insights and tips for researchers and practitioners in both tabular and graph machine learning fields.

**Strengths:**

- The paper provides several useful benchmark datasets for graph based ml.
- The paper provides results across various ml models for tabular learning, graph based models, and combinations of the two.
- The paper provides some recommendations and insights for researchers and practitioners working with tabular and graph data.

**Weaknesses:**

- The paper should address its focus on either transductive or inductive settings. The comparison of the models deal with both models (gbdt and gnns), and the performance gap could be resulting from the fact that gnn models use the unlabeled test data.
- In many real-world situations, the data comes with unseen data. The paper should also address the conversion of relations to graphs and application of gnns in those settings.
- The paper should address address some of the limitations in converting relations into graphs. For instance, the time it takes to preprocess the data and time comparison of the model evaluations.
- The tables of results could be of better quality. (eg, bold-face the best results)

**Questions:**

- On the performance comparison, how does the paper differentiate or address the difference between the transductive and inductive learning methods?
- What are the hyperparameter search space for gradient boosting methods?
- What are some limits in constructing relations into graphs (for example computational limits, etc.,)?
- Even without the relation (graph-like) information, it is advisable to apply construction of graphs with features (for instance Gaussian-kernel) and then apply one of the comparing methods?
- It would be good to have some distinctive markings (ex, bold-face) on the table of results.

---

> ### Author Response · Authors · 2023-11-16
>
> Thank you for the review and valuable suggestions! We address the questions and concerns below.
>
> > The paper should address its focus on either transductive or inductive settings. The comparison of the models deal with both models (gbdt and gnns), and the performance gap could be resulting from the fact that gnn models use the unlabeled test data.
>
> In our paper, we focus on the transductive setting, as it is the most widespread setting in graph ML literature and can be found in many real-world tasks. Indeed, GNNs have an advantage in this setting since they have access to features and incident edges of the test nodes (but not their labels, of course). However, we believe that this advantage is fair, as it is a feature of significantly different design of GNNs and standard tabular models. Also, even standard tabular models like GBDT can potentially leverage unlabeled nodes through pseudo-labeling or other semi-supervised learning techniques. We can extend our experiments with such approaches if required.
>
> > In many real-world situations, the data comes with unseen data. The paper should also address the conversion of relations to graphs and application of gnns in those settings.
>
> This corresponds to the inductive graph ML setup. As we have mentioned above, in our paper, we focus on the transductive setting, which is more popular in graph ML literature. However, applying GNNs to nodes unseen during training is not difficult — we just need to add edges incident to the new nodes in the graph and run the GNN on the updated graph once. Extending the benchmark to the inductive setting would be a natural next step in future research.
>
> > The paper should address address some of the limitations in converting relations into graphs. For instance, the time it takes to preprocess the data and time comparison of the model evaluations.
>
> Please, refer to our response to Reviewer b5gj in Q9.
>
> > The tables of results could be of better quality. (eg, bold-face the best results)
>
> Thanks for this suggestion, we will highlight the best results in our tables.
>
> > What are the hyperparameter search space for gradient boosting methods?
>
> The hyperparameter search space for CatBoost:
>
> | Parameter | Distribution |
> |---|---|
> | ```bagging_temperature``` | $\mathrm{Uniform}[0.0,1.0]$ |
> | ```depth``` | $\mathrm{UniformInt}[3,14]$ |
> | ```l2_leaf_reg``` | $\mathrm{Uniform}[0.1,10.0]$ |
> | ```leaf_estimation_iterations``` | $\mathrm{Uniform}[1,10]$ |
> | ```learning_rate``` | $\mathrm{LogUniform}[0.001,1.0]$ |
>
> The hyperparameter search space for XGBoost:
>
> | Parameter | Distribution |
> |---|---|
> | ```colsample_bytree``` | $\mathrm{Uniform}[0.5,1.0]$ |
> | ```gamma``` | $\{0.0, \mathrm{LogUniform}[0.001,100.0]\}$ |
> | ```lambda``` | $\{0.0, \mathrm{LogUniform}[0.1,10.0]\}$ |
> | ```learning_rate``` | $\mathrm{LogUniform}[0.001,1.0]$ |
> | ```max_depth``` | $\mathrm{UniformInt}[3,14]$ |
> | ```min_child_weight``` | $\mathrm{LogUniform}[0.0001,100.0]$ |
> | ```subsample``` | $\mathrm{Uniform}[0.5,1.0$ |
>
> The hyperparameter search space for LightGBM:
>
> | Parameter | Distribution |
> |---|---|
> | ```feature_fraction``` | $\mathrm{Uniform}[0.5,1.0]$ |
> | ```lambda_l2``` | $\{0.0, \mathrm{LogUniform}[0.1,10.0]\}$ |
> | ```learning_rate``` | $\mathrm{LogUniform}[0.001,1.0]$ |
> | ```num_leaves``` | $\mathrm{UniformInt}[4,768]$ |
> | ```min_sum_hessian_in_leaf``` | $\mathrm{LogUniform}[0.0001,100.0]$ |
> | ```bagging_fraction``` | $\mathrm{Uniform}[0.5,1.0]$
>
> > Even without the relation (graph-like) information, it is advisable to apply construction of graphs with features (for instance Gaussian-kernel) and then apply one of the comparing methods?
>
> Note that directly comparing samples based on their features is not trivial in the case of heterogeneous tabular features (on which we focus in our paper), as these features have widely different scales, distributions, meaning and importance for the task. However, comparing their representation from some layer of a neural network is more meaningful. This is exactly what retrieval-augmented models do (in essence, they simultaneously learn sample representations and a graph based on their similarity). We discuss these models in the related work section and include TabR, a representative example of them, in our experiments. If you know some other meaningful ways to measure the similarity between objects with heterogeneous tabular features for constructing relations between them, please let us know so that we could extend our empirical study.

---

### Official Review · Reviewer_36H3 · 2023-11-03

**Soundness:** 2 fair
**Presentation:** 2 fair
**Contribution:** 2 fair
**Rating:** 3
**Confidence:** 5

**Summary:**

Creates 8 graph datasets with 7-31 features per node and reports result of a number of baseline methods on them.

**Strengths:**

S1. The new datasets may be helpful in that they seem to have useful node-level features. Many of the datasets are created by extending existing graph benchmarks with new features that were originally not provided, which may make these graphs more realistic.

**Weaknesses:**

W1. No insight
W2. No "research" contribution
W3. Language

On W1. While new datasets may be useful, the paper does not provide any tangible insights. First, the datasets are merely described, but there is little rationale about why they have been constructed in this way and what qualitative properties they have. Second, and more importantly, the experimental study reports results of some baseline models using fixed (!) hyperparameters, but does not provide any additional insight beyond pure numbers. I did not learn anything from this study.

On W2. This is more a dataset paper than a research paper. As argued above, the methodology used in dataset construction and, more importantly, empirical study does not enhance our understanding of graph learning.

ON W3. The paper positions itself as graph learning with "tabular" / "heterogenous" features. This raises misleading expectations, as it's really just graphs with a fixed set of vertex-level features.

--- UPDATE AFTER DISCUSSION ---
I'd like to encourage the authors to follow-up on this benchmark, even though I do not think it is ready for publication at ICLR. I raised my overall score to express this point.

The current submission (still) misses a solid discussion of available benchmarks and where they fall short, it is not clear if the submission provides SOTA baselines (which are not "easy" to beat by subsequent work), it's not clear to what extent feature heterogeneity actually plays any role (e.g., would the result change if the original non-heterogenous datasets were used), and the paper's study lacks insight (including insight beyond Platonov 2023).

**Questions:**

None

---

> ### Author Response · Authors · 2023-11-16
>
> Thank you for reading our paper. We reply to the comments below.
>
> > with 6-22 features per node
>
> We would like to clarify that our datasets have from 7 to 31 features, please see Table 1.
>
> > While new datasets may be useful, the paper does not provide any tangible insights.
>
> In Section 5, we discuss the results of our experiments in detail and highlight the main practical insights with separate bullet points (we also briefly summarize them in the introduction). In particular, we note that one should try to convert relational information (which often exists in tabular datasets) into a graph and experiment on the resulting dataset with graph machine learning methods. In that case, standard graph neural networks augmented with embeddings for numerical features tend to provide the best results, while specialized models previously proposed in the literature do not work very well. We believe that these findings are valuable and useful, especially for those who want to understand which machine learning methods actually work in practice.
>
> > First, the datasets are merely described, but there is little rationale about why they have been constructed in this way and what qualitative properties they have.
>
> Our datasets were constructed to cover a wide range of areas and real-world applications where graph-structured data with tabular features appears, and thus to provide a realistic and diverse benchmark. In Section 3.2, we describe in detail various properties of the introduced graph datasets, considering the distribution of node features, various structural properties of graphs, and relations between graph structure and node labels. We also explain why the prediction tasks in our datasets are meaningful and close to real-world applications. If our discussion does not cover some important aspect of the proposed benchmark, and you have any suggestions on how we can improve our analysis, please let us know so that we could include it in the revised version.
>
> > Second, and more importantly, the experimental study reports results of some baseline models using fixed (!) hyperparameters, but does not provide any additional insight beyond pure numbers.
>
> This is not correct. As described in Section 5, for the considered GBDT and tabular DL methods, we have conducted an extensive hyperparameter search using Optuna for each of our datasets. Further, we have chosen a fixed set of hyperparameters in EBBS, which the authors claim to work well across different datasets, and conducted hyperparameter search in BGNN using the original experimental pipeline. As for standard GNNs, they take significantly longer to run, so we conducted preliminary experiments in which we determined a set of hyperparameters with which these models provide strong and stable results across a range of datasets, and used it in all our experiments. Note that one of our results is that GNNs typically outperform graph-agnostic models on our benchmark, thus GNNs tend to produce the best results even with less extensive hyperparameter search.
>
> > This is more a dataset paper than a research paper.
>
> Please note that we have chosen “datasets and benchmarks” as the primary area of our paper. This topic is also listed in the ICLR call for papers, which shows that our paper fits the scope of the conference. We also would like to point out that empirical ML research is impossible without good datasets, thus dataset papers are of great importance for the research community.
>
> > The paper positions itself as graph learning with "tabular" / "heterogenous" features. This raises misleading expectations, as it's really just graphs with a fixed set of vertex-level features.
>
> In our work, we follow the terminology used by previous studies on ML for tabular data in general [1, 2] and graphs with tabular features in particular [3, 4]. In this sense, the proposed graph benchmark represents a collection of tabular datasets with known relations between samples that can be modeled as a graph. If you are aware of other uses of the terms “tabular” or “heterogeneous” in relation to data sample features, please let us know so we could add a discussion of terminology to the paper.
>
> [1] “Revisiting deep learning models for tabular data” in NeurIPS, 2021
>
> [2] “On embeddings for numerical features in tabular deep learning” in NeurIPS, 2022
>
> [3] “Boost then Convolve: Gradient Boosting Meets Graph Neural Networks” in ICLR, 2021
>
> [4] “Does your graph need a confidence boost? Convergent boosted smoothing on graphs with tabular node features” in ICLR, 2022

---

> > ### Comment · Reviewer_36H3 · 2023-11-17
> >
> > Thank you for your response! I'd like to maintain my assessment of this paper w.r.t. to its weaknesses and suitability for ICLR.
> >
> > The proposed benchmarks contains graphs with a fixed set of vertex-level features, some numeric, some categorical. While I generally agree that benchmarking on such data is useful and that the new benchmark dataset may be useful, the study falls itself short.
> >
> > First, the paper does not position itself sufficiently w.r.t. to available datasets or benchmarks, including those that use vertex-level features but don't call it tabular/heterogeneous (W3). A prominent example is the Pokec dataset, which is commonly used in graph learning literature and has the desired features (https://snap.stanford.edu/data/soc-pokec.html). There are >100 node classification benchmarks listed on paperswithcode, there are multiple graph benchmarking papers, and there are studies using graph generators (e.g., GraphWorld). None of these are discussed in this submission.
> >
> > Second, the paper lacks insight. This is to some extent because insufficient arguments for the construction of provided benchmarks (which are presented in a "this-is-what-we-did" style) and how they differ from prior ones. More importantly, the experimental setup does not use a solid and fair setup, esp. w.r.t. to hyperparameter tuning. The authors acknowledge this in their response (and also state in in Appendix A). In particular, using "some" (manually set) fixed setting for some methods, originally provided hyperparameters for other methods, and an extensive search for yet other methods does not provide for a convincing setup. Also, the method lacks recent baselines/methods (e.g., GloGCN [A] and its predecessor Link-X appear natural, but there are also many other recent methods).
> >
> > [A] Li at al., Finding Global Homophily in Graph Neural Networks When Meeting Heterophily. ICML 22

---

> > > ### Author Response · Authors · 2023-11-17
> > >
> > > Thanks for your quick response, we appreciate your engagement in the discussion. Let us briefly reply to the additional questions raised.
> > >
> > > > A prominent example is the Pokec dataset, which is commonly used in graph learning literature and has the desired features (https://snap.stanford.edu/data/soc-pokec.html). There are >100 node classification benchmarks listed on paperswithcode, there are multiple graph benchmarking papers, and there are studies using graph generators (e.g., GraphWorld).
> > >
> > > In fact, before conducting our study, we have carefully examined numerous graph datasets that can be found in open access and filtered them due to various reasons — most of them contain a significant amount of NaN values (e.g., the mentioned Pokec dataset), have very few (if any) tabular features (e.g., OGB and many other popular datasets, as we discuss in the related work section), have issues with formulating a meaningful prediction task, represent synthetically generated data rather than real-world graphs (e.g., the mentioned GraphWorld benchmark has synthetically generated homogeneous features), etc. The remaining datasets that are suitable for our purposes we include in adapt and include to our benchmark. Since there are only a few datasets, we had to collect new graph datasets that meet our requirements: provide meaningful tasks, contain heterogeneous tabular features, and have diverse graph structures.
> > >
> > > > More importantly, the experimental setup does not use a solid and fair setup, esp. w.r.t. to hyperparameter tuning. The authors acknowledge this in their response (and also state in in Appendix A). In particular, using "some" (manually set) fixed setting for some methods, originally provided hyperparameters for other methods, and an extensive search for yet other methods does not provide for a convincing setup.
> > >
> > > Our approach to selecting hyperparameters is determined by how many resources it takes to conduct one experiment with a given method — for those methods that can be trained and applied very quickly (e.g., GBDT and tabular deep learning models), we have conducted extensive hyperparameter search; for those methods which require much more time to set up an experiment (e.g., standard GNNs and their combination with GBDT), we decided to rely on the hyperparameter search performed in the official implementation or use those hyperparameters that have proved to be suitable across many graph datasets and prediction tasks in the preliminary experiments. We consider such an approach to be quite fair and reasonable. Note that one of our key takeaways is that graph-agnostic methods (for which we perform extensive hyperparameter search) are outperformed by GNNs (for which we consider fixed hyperparameters), thus adding more extensive hyperparameter search to GNNs will not change our conclusions.
> > >
> > > > Also, the method lacks recent baselines/methods (e.g., GloGCN [A] and its predecessor Link-X appear natural, but there are also many other recent methods).
> > >
> > > Note that we have evaluated a representative set of popular GNNs (6 models). Importantly, as shown in [1], these models typically outperform many recently proposed methods, such as the mentioned GloGNN. Further, we would like to point out that GNNs achieve the best results on our benchmark. Thus, adding even more GNN models to the comparison will not change our conclusions.
> > >
> > > [1] “A critical look at the evaluation of GNNs under heterophily: are we really making progress?” in ICLR, 2023

---

### Comment · Area_Chair_69UV · 2023-11-20
**reviewers, please acknowledge the responses from the authors**

Dear reviewers: Please read the replies from the authors carefully, and submit your reactions. Please be open-minded in deciding whether to change your scores for the submission, taking into account the explanations and additional results provided by the authors.

Thank you!

---

### Author Response · Authors · 2023-11-22
**Revised paper**

Following the suggestions of some reviewers, we have extended the discussion of such aspects of our graph benchmark as the choice of threshold in bipartite networks, construction of graphs using external relational information instead of comparing nodes based on their heterogeneous tabular features, and time required to construct graphs. Please, refer to Appendix A for this discussion. Moreover, we have marked best results in our tables and added details on hyperparameter search for tabular classic and deep learning models in Appendix D.

We thank the reviewers for their feedback and valuable suggestions. We also hope that we have managed to address most concerns raised during the rebuttal.

---

### Meta-Review · Area_Chair_69UV · 2023-12-14

**Metareview:**

Three reviewers remain negative after detailed discussion with the authors. One reviewer is positive, but is less experienced.

The topic “datasets and benchmarks” is in scope for ICLR, but this submission requires further work to meet the high ICLR standard.

An additional reference for the authors: arXiv:2305.15843 Submitted on 25 May 2023, "TabGSL: Graph Structure Learning for Tabular Data Prediction" by Jay Chiehen Liao, Cheng-Te Li.

**Justification For Why Not Higher Score:**

The research contributions are limited.

**Justification For Why Not Lower Score:**

The topic is important, as is the general idea of using the links between samples in tabular data.

---

### Decision · Program_Chairs · 2024-01-16

Reject